# The principles of recovery-oriented mental health services: A review of the guidelines from five different countries for developing a protocol to be implemented in Yogyakarta, Indonesia

M. A. Subandi[1]*, Maryama Nihayah[1], Carla R. Marchira[2], Trihayuning Tyas[1], Ariana Marastuti[1], Ratri Pratiwi[3], Fiddina Mediola[4], Yohanes K. Herdiyanto[5], Osi Kusuma Sari[6], Mary-Jo D. Good[7], Byron J. Good[7]

1 Faculty of Psychology, Gadjah Mada University, Yogyakarta, Indonesia, 2 Department of Psychiatry, Faculty of Medicine, Public Health and Nursing, Gadjah Mada University, Yogyakarta, Indonesia, 3 Faculty of Psychology, Mercu Buana University, Yogyakarta, Indonesia, 4 Puri Nirmala Special Hospital, Yogyakarta, Indonesia, 5 Department of Psychology, Faculty of Medicine, Udayana University, Denpasar, Bali, Indonesia, 6 Directorate of Mental Health, Ministry of Health, Jakarta, Indonesia, 7 Department of Global Health and Social Medicine, Harvard Medical School, Harvard University, Boston, Massachusetts, United States of America

* subandi@ugm.ac.id

**Data Availability Statement:** All relevant data are within the paper.

## Abstract

### Background

Recovery-oriented mental health service has become the focus of global change in mental health services. Most of North industrialized countries have adopted and implemented this paradigm in the last two decades. Only recently that some developing countries are trying to follow this step. In Indonesia's case, there has been little attention to developing a recovery orientation by mental health authorities. The aim of this article is to synthesize and analyze the recovery-oriented guidelines from five industrialized countries that we can use as a primary model for developing a protocol to be implemented in community health centre in Kulonprogo District, Yogyakarta, Indonesia.

### Method

We used a narrative literature review by searching for guidelines from many different sources. We found 57 guidelines, but only 13 from five countries met the criteria, including five guidelines from Australia, one from Ireland, three from Canada, two from the UK, and two from the US. To analyze the data, we used an inductive thematic analysis to explore the themes of each principle as described by the guideline.

### Result

The results of the thematic analysis revealed seven recovery principles, including (1) cultivating positive hope, (2) establishing partnerships and collaboration, (3) ensuring

**Funding:** This study was funded by Innovative and Productive Research Program, the Minister of Finance, the Government of Indonesia The authors who received the award: MAS CM TT AM are Grant numbers: 110/LPDP/2019 URL funder: https://lpdp.kemenkeu.go.id/riset/kebijakan-rispros-umum/ The funders had no role in study design, data collection and analysis, decision to publish, or preparation of the manuscript.

**Competing interests:** The authors have declared that no competing interests exist.

organizational commitment and evaluation, (4) recognizing the consumer's rights, (5) focusing on person-centeredness and empowerment, (6) recognizing an individual's uniqueness and social context, and (7) facilitating social support,. These seven principles are not independent, rather they are interrelated and overlap each other.

## Conclusion

The principle of person-centeredness and empowerment is central to the recovery-oriented mental health system, while the principle of hope is also essential to embracing all the other principles. We will adjust and implement the result of the review in our project focusing on developing recovery-oriented mental health service in the community health center in Yogyakarta, Indonesia. We hope that this framework will be adopted by the central government in Indonesia and other developing countries.

## Introduction

Recovery-oriented mental health services can be traced to the 1970s consumer movement in the US [1]. Initially, the movement tried to change psychiatric approaches to mental illness, notably by challenging the pessimistic idea that schizophrenia is incurable. Supported by psychiatrists who were also skeptical of this idea, a number of consumers wrote about their personal experiences. A large number of personal accounts have since appeared in well-respected scientific journals, such as *Schizophrenia Bulletin*, *Psychiatric Services*, *Psychiatric Rehabilitation Journal*, and *Psychiatric Rehabilitation Skill*. Spaniol and Koehler [2] then compiled these accounts into an anthology titled *The Experience of Recovery*. These accounts have become the founding stories of the recovery movement. They have provided a starting point of advocacy for health services to go beyond symptom reduction by promoting a meaningful life in the community.

The consumer movement had at least three different agendas. First, it fought against stigma and sought to change the public perception of mental illness. Second, since the movement highlighted the potential harm that the mental health profession could cause, it actively sought to change professional practice. Third, it played an influential role in shaping government policies on mental health care systems [3–5].

Anthony [3] introduced the notion of a recovery-orientation for mental health services, and in 2002, the state government of Connecticut adopted a policy for promoting a recovery-oriented system of care. Since then, the recovery movement and recovery-oriented mental health services have garnered global interest. Although it is not derived from an evidence-based research of new psychiatric medications or an accumulating body of research on clinical improvements, a recovery orientation has become part of the transformation of mental health systems [6]. In the last two decades, a commitment to recovery has become a major orientation in mental health policy, guidelines, action plans, and practice at the international level [7].

Pincus et al [8] reviewed the implementation of recovery-oriented mental health services from selected industrialized countries. In Australia, recovery became an important priority for its national and state mental health policies. In 2013, the Australian Health Ministers' Advisory Council released the National Framework for recovery-oriented mental services. Some of the guidelines has also been released by many different institutions, such as St. Vincent's Hospital

and the West Australian Association for Mental Health. In Canada, the recovery idea has been incorporated into mental health services at the state/territorial level since 2010. In this regard, the Healthy Minds, Healthy People Initiative is a 10-year plan which promotes a recovery approach to transform the mental health system in Canada. The recovery movement has also been adopted in the UK as the guiding vision of its government policy since 2001. In Ireland, the Department of Health and Children recommended that Irish mental health services adopt a recovery perspective in 2006.

Following in their footsteps, other developed European countries, such as Scotland, Norway, Germany, the Netherlands, and Italy, have also incorporated this orientation into their healthcare services. Later, several non-European countries, such as Israel [9], Hong Kong [10], and South Africa [11, 12], also adopted this approach to care. Recently, Chang et al. [13] provided a report of recovery-oriented mental health services in Taiwan.

In Indonesia's case, there has been little attention to developing a recovery orientation by mental health authorities. However, we note two projects that have been trying to develop a recovery framework. Stratford et al. [14] reported a community recovery rehabilitation project in the Sukabumi district of the West Java province. This project was under the Ministry of Social Affairs and was claimed to be successful. However, its sustainability was later questioned. The West Java mental hospital initiated a hospital-based recovery-oriented program called "Kampung Walagri" [15] recently. It is, however, unclear whether these two projects are connected.

Since 2012, our research team has been trying to strengthen mental health programs in community health centers (*Puskesmas*) in one district through a research collaboration between Harvard University and Gadjah Mada University [16–18]. Titled *"Bangkit Jiwa"* (mental revival), our project—particularly over the last two years—has concentrated on adopting a recovery-oriented mental health approach by collaborating with the local government, particularly the district health office, of the Kulonprogo district, in the special province of Yogyakarta, Indonesia. After much deliberation, the local government agreed to incorporate a recovery-oriented program into its five year mental health plan.

As the initial step of the project, this literature review aims to synthesize the guidelines of various recovery-oriented mental health services in the world. The aim of this review is to guide the direction of this entire project.

## Methods

This literature review uses a narrative analysis method. We searched for the relevant literature from various sources, such as Google Scholar, the online archive of the National Institute for Health and Care Excellence (NICE) UK, the US Substance Abuse and Mental Health Service Administration (SAMSHA) and Science Direct. We used several keywords. such as 'recovery' AND 'guideline' AND 'mental health'. Alternatively, we also used the keywords 'practical guide to recovery' AND 'mental health'. We did not use formal search engines, such as Scopus, EBSCO, or ProQuest, since they mostly search for journal articles that do not have practical guidelines.

The selection criteria for these guidelines/articles are those that are available online, can be downloaded, and are written in English. Other criteria include the availability of a definition for recovery-oriented services and a description of the service principles used. From the search, we found 57 guidelines. Of these guidelines, 13 met the criteria for further analysis. These 13 guidelines were from five different countries: five guidelines from Australia, one from Ireland, three from Canada, two from the UK, and two from the US. The link of guidelines can be seen in the References number 20–32.

To analyze the data, we used an inductive thematic analysis to explore the themes of each principle as described by the guideline. An inductive thematic analysis is the process of coding data without trying to fit it into a pre-existing coding framework or the researcher's analytic preconceptions. Thematic analysis is used because it allows a flexible and pragmatic approach that can provide rich and detailed data [19].

The analysis was first carried out simultaneously by members of this research team to find important notes and appropriate themes. From the themes found, discussions were held regarding the category of themes and descriptions. Furthermore, to get a theme that describes the findings more accurately and comprehensively, a joint theme review was conducted. To reduce the possible errors, a senior professional checked and evaluated the result of analysis as a quality control. Technically, the steps taken in this stage are 1) familiarizing ourselves with the data by reading it over and over again, 2) creation of the initial code, drawing on initial notes, 3) searching for a theme, 4) doing a theme review, 5) defining and naming the theme, and 6) conducting a follow-up analysis by calculating the frequency of keywords of themes in each document.

Ethical approval for the project was obtained from the Ethical Committee, Faculty of Psychology, Gadjah Mada University, Yogyakarta, Indonesia number 4628/UN1/FPSi.1.3/ SD/PT.01.04/202. The Ethic committee waived the need for consent because this is a review study. In addition, all the documents that we used for data analysis are available online.

## Results

A total of 135 recovery principles with 153 units of explanation were obtained from 13 documents. Each unit of explanation varied in length, from one sentence to a paragraph.

Table 1 shows the list of guidelines as the primary source of data in this study.

From the results of the thematic analysis, we synthesized seven principle themes listed according to the frequency of emergent keywords of the themes in the guidelines: (1) Cultivating positive hope, (2) Establishing partnership and collaboration, (3) Ensuring organizational commitment and evaluation, (4) Recognizing the consumer's human and civil rights, (5) Focusing on person-centeredness and empowerment, (6) Recognizing an individual's uniqueness and cultural context, (7) Facilitating social support. Table 2 shows the summary of the principles and the frequency of keywords of themes used in each guideline. The description of each theme follows the table.

### 1) Cultivating positive hope

Hope is the theme that appears most frequently in the guidelines, despite no word of 'hope' in guideline D [23]. Meanwhile, guideline I [28] includes the word 'hope' in the title and clearly states that recovery is about hope. Guideline C3 [22] mentions the first domain of recovery is hope.

*Hope is central to recovery. . .promoting a culture and language of hope and optimism*

[22: p.5]

*The emotional essence of recovery is hope, a promise that things can and do change, that today is not the way it will always be*

[23: p.12].

**Table 1. List of guideline.**

| Country | Guideline Identification | Reference Number | Year | Guidelines Title | Publisher |
|---|---|---|---|---|---|
| Australia | A | 20 | 2011 | Framework for Recovery-oriented Practice | State Government Victoria |
| | B | 21 | 2012 | The Framework for Recovery-oriented Rehabilitation in Mental Health Care | Government of South Australia |
| | C | 22 | 2013 | A National Framework for Recovery-oriented Mental Health Services. A Guide for Practitioners and Providers | Commonwealth of Australia |
| | D | 23 | 2015 | Rethink Mental Health: A Long-term Plan for Mental Health in Tasmania 2015–2025 | Department of Health and Human Services. Government of Tasmania |
| | E | 24 | 2017 | National Strategic Framework for Aboriginal and Torres Strait Islander Peoples' Mental Health and Social and Emotional Wellbeing 2017–2020 | Government of Australia |
| Ireland | F | 25 | 2012 | Advancing Community Mental Health Services in Ireland—Guidance Papers | "A Vision for Change" Group |
| Canada | G | 26 | 2009 | Toward Recovery & Well-Being: A Framework for a Mental Health Strategy for Canada | Mental Health Commission of Canada |
| | H | 27 | 2018 | Best Advice: Recovery-Oriented Mental Health and Addiction Care in the Consumer's Medical Home | The College of Family Physicians of Canada |
| | I | 28 | 2015 | Guidelines for Recovery-Oriented Practice: Hope, Dignity, Inclusion. | Mental Health Commission of Canada. |
| UK | J | 29 | 2008 | Making Recovery a Reality | Sainsbury Centre for Mental Health |
| | K | 30 | 2014 | REFOCUS: Promoting Recovery in Community Mental Health Services (Rethink recovery series: volume 4, 2nd issue) | Institute of Psychiatry, King's College University, University of London |
| USA | L | 31 | 2009 | Guiding Principles and Elements of Recovery-Oriented Systems of Care: What do we know from the research? | Substance Abuse and Mental Health Services Administration (SAMHSA) |
| | M | 32 | 2008 | Practice Guidelines for Recovery-Oriented Care for Mental Health and Substance Use Conditions | Connecticut Department of Mental Health and Addiction Services |

**Table 2. Summary of the principles in each guideline.**

| Guideline Reference Number | Principles | | | | | | |
|---|---|---|---|---|---|---|---|
| | (1) Cultivating Hope | (2) Partnerships & collaboration | (3) Commitment & Evaluation | (4) Human & civil rights | (5) Person-centered & empowerment | (6) Uniqueness & cultural Context | (7) Social Support |
| A [20] | 22 | 17 | 2 | 22 | 3 | 0 | 0 |
| B [21] | 14 | 55 | 21 | 20 | 11 | 3 | 5 |
| C [22] | 37 | 53 | 28 | 34 | 18 | 0 | 4 |
| D [23] | 0 | 15 | 29 | 7 | 14 | 0 | 1 |
| E [24] | 1 | 13 | 10 | 13 | 14 | 0 | 2 |
| F [25] | 7 | 13 | 6 | 0 | 3 | 0 | 16 |
| G [26] | 30 | 7 | 12 | 18 | 11 | 2 | 1 |
| H [27] | 11 | 5 | 6 | 2 | 2 | 1 | 0 |
| I [28] | 112 | 73 | 22 | 44 | 26 | 3 | 2 |
| J [29] | 24 | 2 | 7 | 3 | 2 | 0 | 0 |
| K [30] | 11 | 0 | 5 | 0 | 5 | 1 | 3 |
| L [31] | 18 | 15 | 21 | 1 | 10 | 1 | 14 |
| M [32] | 29 | 22 | 14 | 23 | 44 | 2 | 2 |
| TOTAL | 316 | 290 | 187 | 187 | 161 | 68 | 50 |

Guideline I [28] suggests that the first step in the recovery journey is cultivating hope. Not only hope among consumers but also among service providers. A sense of hope could be achieved by broadening the concept of recovery. Almost all guidelines suggest that recovery does not mean eradicating symptoms and a full return to premorbid functioning. Instead, recovery focuses more on social functioning, where people with mental illness can achieve a meaningful life, with or without symptoms.

The main task of service providers is to help consumers accept their illness and have positive expectations, be optimistic, and believe in the opportunity to grow and develop [28]. With this hope, consumers can construct a positive self-identity, while having an open view of themselves and their environment. In this regard, even a positive language from the service providers can make consumers feel valued [27, 28].

The belief that recovery is real provides the essential and motivating message of a better future; that people can and do overcome the internal and external challenges, barriers, and obstacles that confront them, such as stigma and discrimination [20–22, 24]. To ensure this, a consumer's hopes must be internalized and fostered by their families, peers, providers, allies, and others.

## 2) Establishing partnerships and collaborations

The theme of establishing partnerships and collaborations consists of two keywords, 'partnership' and 'collaborations'. Some guidelines use these two words separately [23, 25, 28], but other guidelines combine them as ' collaborative partnership' [20, 28, 32]. The journey of recovery cannot be accomplished alone. It needs a true working partnership. Most guideline suggest that recovery-oriented mental health service needs partnership and collaboration between consumers, their families, and service providers.

*Mental health care is responsive to the range of different needs people may have, which involves effective collaboration with non-mental health service providers*

[20: p.15].

*. . .to initiate collaborative partnerships and coordinated planning efforts that include government policy planners, nongovernmental organizations, community agencies, people with lived experience and family caregivers*

[28: p.96]

Most of the guidelines suggest that recovery-oriented mental health services should recognize the contribution of many different parties. Hence, partnerships are vital between consumers, caregivers, families, communities, health care staff, as well as multi-disciplinary service providers [23–25, 28], governments, organizations, and communities [21], key agencies and voluntary groups in the community, as well as mainstream health services (e.g., primary care teams and other referring agents); social welfare; education services, and housing authorities [25]. Moreover, these stakeholders should work together in a respectful collaboration to support the recovery process. Out of these, the consumer's involvement must be acknowledged primarily. They are considered experts of their own lives and experiences [21, 29] while mental health services are considered experts when it comes to providing the treatment. The helping relationship between clinicians and consumers should be changed from being "expert-consumer" to being "coaches" or "partners" on a journey of discovery [28]. Service providers need to share power and acknowledge the contribution of "experts-by-experience" [22, 29]. They should "do with" the consumers, not "do to" or "do for" them [21, 30, 32]. Moreover, a

consumer's peers can also support their recovery through formal self-help, informal encounters, mutual assistance, and exposure to their stories of recovery [24, 30, 31].

Here, the partnership should also include coordination and collaboration with a range of relevant agencies beyond the mental health system, including health services, disability services, employment, education, training services, and housing services [21, 22, 25]. They serve as referring agencies and explore new service partnerships. Moreover, partnerships that provide community support to aid social inclusion is also essential [22, 23, 27, 30, 32]. The involvement of various parties and agencies can enable consumers to maximize their potential, achieve wellbeing, and a positive future. Various agencies, NGOs, and other relevant stakeholders should be able to provide integrated, innovative, and flexible services so that they can effectively respond to the needs of the consumers, families, or communities [21, 22].

## 3) Ensuring organizational commitment and evaluation

Table 1 shows that the frequency of keywords 'commitment' and 'evaluation' used in the guidelines is relatively high, meaning that it is an essential aspect of recovery-oriented services. All of the guidelines mention these keywords. A recovery-oriented services are not only about how services are provided, but also about how they are committed to facilitating a recovery approach using organizational resources. The recovery values should be implemented in all management processes, such as recruitment, professional development, supervision, appraisal, audit, service planning, and operational policies [G1, G3, G9, G12, G13]. The appropriate recovery language should also be incorporated into all key organizational documents and publications [21, 22, 29, 31, 32].

Almost all of the documents suggest that the attitude of the staff throughout the organization is very important in shaping environments which facilitate recovery, especially those that support people in developing and implementing their own recovery plans.

*Service and work environments and an organizational culture that are conducive to recovery and to building a workforce that is appropriately skilled, equipped, supported and resourced for recovery-oriented practice*

[22: p.4].

It is also important that an organization be committed to involving consumers in its functioning at all levels. By doing this, the organization could have a big impact in the service provided. A professional consumer (a former consumer with an academic background such as psychiatrist, clinical psychologist or social worker) with "lived experience" could work alongside other professionals in their organization [21, 22]. They not only serve as a model figure for people in the journey toward recovery, but can also contribute to service planning and evaluation activities. It is also important that the organization commits to fostering a culture of continuous service improvement by implementing evaluations to ensure best practices and a high-quality service system [20–23, 28, 29].

*Promote research and evaluation activity that involves peers and people in recovery; incorporate findings in service improvements and standards of practice*

[28: p.87]

## 4) Recognizing the consumer's human and civil rights

The keywords 'human rights' and 'civil rights' are in the same position as the organisational commitment and evaluation theme. Almost all of the documents mention this theme. Historically, the recovery movement began as a civil rights movement aimed at restoring consumers' human rights and their opportunity to community inclusion and social integration [22, 23, 28, 29, 32]. These values have been incorporated in the guiding principle of recovery. Service providers should respect the rights and dignity of the consumers by focusing on their strengths and abilities [26–28]. Moreover, recovery-oriented mental health services must uphold, support, and protect a consumer's legal, citizenship and human rights [2–4]. This is an important goal to be accomplished in the practice of recovery-oriented services.

Services providers should also give equal opportunities for every consumer to participate in education, work, as well as the community to encourage social inclusion [22, 23, 26, 31, 32]. According to this principle, they should remove the barriers to stigma and discrimination because they can negatively impact the consumer's recovery and well-being. Consumers are supported to have the same rights as others in the fields of education, employment, community involvement, and opportunity to make choices, and achieve wellbeing.

*building healthy public policy (such as reducing stigma, facilitating social inclusion, upholding human rights, facilitating employment*

[23: p.17]

## 5) Focusing on person-centeredness and empowerment

The keywords 'person-centered' and 'empowerment' are integrated into one theme because they focus on the consumers. Most of the documents suggest the importance of holistically seeing the person in recovery and acknowledging their power or autonomy and empowerment [20–26, 28, 30, 32]. In the first aspect, a person with mental illness needs to be seen holistically, i.e., their life and circumstances. This is as not limited to the person's mental health, but also includes their physical health, emotions, developmental stage, and gender. It also includes external aspects such as cultural, social, and economic aspects. The providers should understand the individual's complex needs and aspirations and view their personal recovery as the primary process of working towards wellness.

The principle of empowerment indicates that the service providers acknowledge the autonomy of the consumers. Recovery is not necessarily about cure but is about having opportunities for choices, living a meaningful, satisfying and purposeful life, and being a valued member of the community [22, 23, 25–28, 30, 31]. Service providers should believe that their consumers have the ability and right to make their own life decisions. In brief, the consumers are at the heart of the recovery-oriented culture. They should describe their own experiences and journeys and affirm their personal identity beyond the constraints of their diagnoses. They have the opportunity to lead, control, and select the service processes and practices that support their recovery. Consumers are active agents of change in their lives and not passive recipients of services [22, 29–32]. They deserve to be included in the development of a recovery plan. In this regard, all services must be structured to support individuals in building their strengths and taking responsibility for their lives, while understanding that they may need support from time to time.

The principle of taking a person-centered orientation refers to the view that recovery is built on honoring each person's capacities, knowledge, talents, coping abilities, resources,

and potential for growth [25, 30, 32]. Hence, the primary focus should be put on a person's strengths and abilities, not on their weakness and disability.

> promoting self-determination and dignity, adopting a holistic and strengths-based approach, fostering hope and purpose and sustaining meaningful relationships, also form the foundation of a recovery orientation

[28: p.17]

## 6) Recognizing an individual's uniqueness and cultural context

Most the documents recognise that the process of recovery is a personal, unique and complex journey [21, 22, 28, 29, 32]. There is no two people will have identical paths or use the same benchmarks to measure their journeys. Therefore, recovery-oriented services and care should be tailored according to an individual's strengths. Service providers should evaluate each individual's needs, including the physical, social, emotional, cultural and spiritual aspects of their life. The focus should be placed on the development of individualized self-management plans rather than compliance with a standard treatment regime [28, 31, 32]. Here, personal experience is central and vital to mobilizing the individual's own resources. It involves being sensitive to and respectful of each individual, with regards to their values, beliefs and culture. This also includes their human rights.

The context in which the consumers live, including their social, economic, and cultural background, will influence on the recovery process [20–22, 26, 28]. Therefore, it is important to consider and utilize these values as a source of support for the recovery process. Recovery considers how the cultural and spiritual aspects of consumers lives impact on their view of mental health and the process of treatment. Given this context, traditional care or other culturally relevant practices should also be considered, as these practices have a sense of the individualism and collectivism of their culture.

> *While each person's journey of recovery is unique, people do not journey alone; their journeys take place within a social, familial, political, economic, cultural and spiritual context that impacts their mental health and well-being*

[28: p.11].

## 7) Facilitating social support

Some guideline suggest that supporting a consumer's inner circle or community, such as their family, relatives, peers, and other supportive people, play a vital role in the recovery process [23–25, 28, 30]. A supportive community is a natural resource that encourages the creation of a positive self-identity and hope for recovery through social inclusion and community participation. Service providers should support and encourage individuals to remain connected with the significant people in their life. A consumer should have someone whom they can trust to "be there" in times of need.

In this regard, service providers can leverage support groups with the consumers and their immediate environment. It is important to note that service providers are required to have prior adequate knowledge of the local community, including identifying potential barriers that may exist. Factors like stigma, discrimination, gendered norms, and negative societal perceptions of mental illness could hinder the recovery process [20, 22, 23, 26, 28, 30–32].

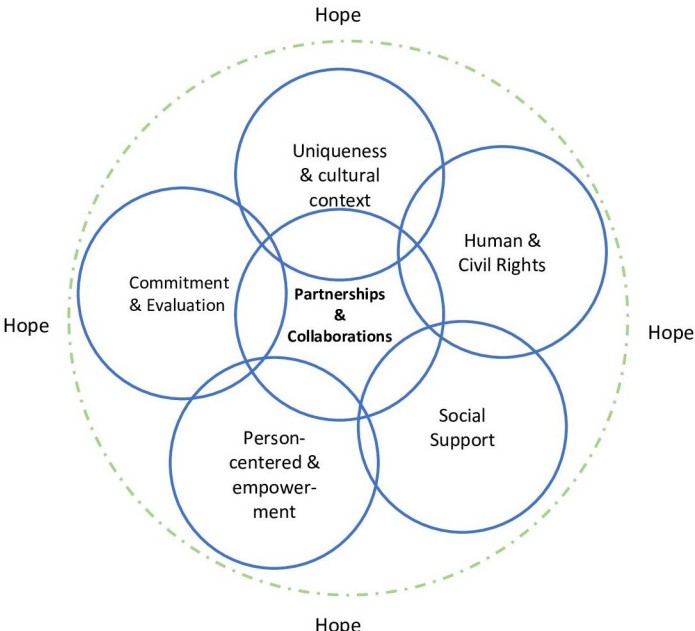

**Fig 1. Interrelated principles of recovery-oriented mental health service.**

*Strong connections among people are the foundation of mental health and wellbeing and resilience for individuals, families and the broader community. These connections help reduce stigma, nurture social inclusion and respect diversity*

[23: p.13].

*Family and other supporters are often crucial to recovery and they should be included as partners wherever possible. However, peer support is central for many people in their recovery*

[29: p.0].

The seven principles described above are interrelated. Fig 1 shows that the principles overlap each other. In the center of the circle is the principle of partnerships and collaborations. All the other principles, except hope, surround the center while overlapping with each other, indicating their integration. Meanwhile, the principle of hope is located outside the circle, indicating that it is essential to embracing all the other principles.

## Discussion

This study aimed to identify the principle of recovery-oriented mental health services that have been included in the guidelines from five countries. The thematic analysis identified seven essential principles of recovery-oriented mental health services that have been included in the guidelines from five countries. Following the aim of this study, we will use these principles to develop a protocol and implement it in the context of our projects in Yogyakarta, Indonesia.

Most of the principles align with the principles described in several studies. Several studies suggested that hope is a central principle of recovery [33–35]. In the context of mental health recovery, hope is defined as a future-oriented expectation to achieve goals and personally

valuable relationships [35]. Hope means remembering that recovery can be a long-term process with many setbacks and advances along the way [33] and non-linear and complex [7], but still believing that recovery is a reality. In the process of seeking recovery, hope is a word combined with other factors that promote recovery.

Hope is a very subtle concept, but serves as a "guiding principle" in recovery [36]. Thus, this simple belief belongs not only to the consumers themselves, but also professional workers. By embracing the possibility of recovery, mental health service providers could cultivate their own hope. Later, they could share their hope to the consumer and their family. Hope has enormous practical consequences and was found to have a positive impact on the recovery process [37–39]. It has the power to inspire change in everyone. Hope is the core of recovery-oriented services.

There are several ways to build and maintain hope. The experience of regaining authority through self-empowerment with adequate environmental support is essential for rebuilding and maintaining the hope of recovery [40]. Hope can also be cultivated through a welcoming and accepting environment. The humanistic principle of non-judgmental listening, genuineness, and warmth can facilitate personal growth. This also means focusing on the strengths and positive outcomes rather than deficits. Consumers could learn to reframe their illness-related setbacks as part of the long-term process towards achieving recovery goals. Peer support from fellow users or survivors has also been found to promote hope, self-determination, participation in services, along with the knowledge of lived experiences to help each other more than professionals can [39, 41, 42]. In this regard, several psychosocial interventions have been recommended to support consumers and foster hope, including advance directives [43] Wellness Recovery Action Planning (WRAP) models [42, 44] and Illness Management programs and Recovery (IMR) [45].

Another strategy to cultivate hope is using a positive language as suggested by two guidelines [27, 28]. Service providers should communicate hopeful messages about recovery, believing that recovery—in its broader definition of being able to live a meaningful life—is a reality. This hopeful message and optimism are essential for maintaining a focus on strengths, building resources, and helping people sustain relationships. In the history of the recovery movement, activists changed the word "patient" into "consumer" (one who uses mental health facilities). In the UK, the word "users" is preferred. To adopt this principle in our Kulonprogo, Indonesia project, we use the term *"penjiwa" (penyintas jiwa)* meaning "a survivor of mental problem problems". This term conveys the meaning of hope and resiliency. Meanwhile, the formal term is *ODGJ (Orang Dengan Gangguan Jiwa)* meaning "people with a mental disorder". We consider this term to be stigmatizing since it includes the word "disorder". This is why we have chosen to emphasize n the word "survivor".

This research found the complexity of recovery-oriented mental health services, including personal, clinical, organizational, and community issues. This is in line with the narrative of mental health recovery found by Llewellyn-Beardsley et al. [7]. These authors described the characteristics of mental health recovery as being diverse and multidimensional, which incorporated social, political, and right aspects. Therefore, establishing partnerships and collaborations become essential principles.

The principle of 'ensuring organisation commitment and evaluation' is essential because developing a focus on recovery-oriented involves transformation within mental health systems. Slade et al. [42] suggested that human systems do not easily transform. Integrating many different perspectives and implementing them into practice requires a long process. Furthermore, Slade et al. [42] identified three scientific challenges of recovery-oriented mental health, including broadening cultural understandings of recovery, implementing organizational transformation, and promoting citizenship.

Several guidelines mention the essential aspect of principle 'recognising the human and civil right' in recovery-oriented mental health service [22, 23, 28, 29, 32]. Davidson et al. [6] reviewed the history of recovery movement and they suggested that this movement was rooted in the civil rights movement of the 1950s and 1960s and the independent living and disability rights movement of the 1970s in the US. They also suggested that to achieve recovery, a consumer should have "...*the sense of being a full citizen and have certain rights (e.g., the right to community inclusion) and resources (e.g., a home, an income) and be able to take on certain roles and responsibilities (e.g., neighbor, voter) while having meaningful relationships with others that offer the person a sense of belonging*" [6].

Focusing on person-centered and empowerment is another feature of recovery-based mental health services [7, 33, 46]. It relates considers recovery as a consumer's subjective experience at its core. Mental health providers should recognize that each consumer has their own knowledge, strengths, abilities, and backgrounds. This is because the knowledge related to narratives or experiences of the consumer's recovery process can increase empathy and understanding among health staff [47], act as a peer support mechanism [48], and offer important clues on how consumers can be facilitated in the recovery process [49] and guide clinical intervention and evaluation strategies [34]. However, accepting consumer diversity can be challenging. In their study of mental health providers, Lodge et al. [50] found that the ability of health workers to adapt interventions to the needs of different consumers is still insufficient due to their lack of knowledge, clarity, and experience in applying this principle. Another challenge is noting and guiding the mental health provider's changing perception of the consumer as the object of care into the subject of treatment [51]. In this regard, training can be an important mechanism for enhancing skills and promoting recovery-oriented practices [52].

Empowerment is another issue that needs to be considered. The main difference between traditional mental health services and the more recovery-oriented ones is reflected in the principle of empowerment. In the former, consumers are seen as passive recipients of the treatment practice, while in the latter, the consumer must be involved in the preparation and planning of care [53]. They are given the opportunity to express their goals and self-direct their care. Empowerment can be seen further in the idea of regarding consumers as "experts-by-experience", while professionals are seen as "experts in treatment availability" [26, 27]. According to this principle, the consumer has the right to be involved in the system. It is in line with the principle of partnership. The providers not only collaborate with other government institutions and communities, but also with the family and the consumer's organization.

Considering the consumer's uniqueness and cultural characteristics is another important principle of recovery. The narratives of the recovery process are diverse and multidimensional, even involving aspects of social, cultural, political, and citizenship rights [7, 54]. The consumer's internal world interacts with systemic socio-political forces that affect their health and well-being, making these factors important when focusing on consumer independence [55]. In our early Indonesian project, we used a culturally specific term for the title of our project (*Bangkit Jiwa*). This term was taken from our previous project [56] because it has several positive meanings which are in line with the concept of recovery. The word *"bangkit"* represents the idea of gaining insight and awareness, acquiring the motivation to change, and changing from being passive to being active. "By conceptualizing the recovery process in terms of *'bangkit'*, participants appropriated a powerful set of cultural meanings that exerted a transformative effect on their lives and their approach to their illnesses" [56].

The complexity of recovery-oriented mental health services found in this study align with a number of challenges found in the literature, especially regarding the practical application of the recovery principles and concepts as services. Shera and Ramon [57] compared the implementation of this orientation at multiple levels of practice in England and Canada.

They identified similar challenges between the two countries. In the practical level, the challenge includes definitional clarity, stigma as the primary hindrance, and resources of recovery. It is not an easy task to change the definition of recovery from a medical (focusing on symptom reduction) perspective to one that is more consumer-driven perspective where recovery is seen as a personal journey that is unique to each individual [58]. At the policy level, the challenges include policy and program implementation and political will. Shera and Ramon [57] found that from an organizational, a cost-effective consideration is very important in reforming the current system of practice. This is the reason why changes in the mental health systems usually do not have a strong public support and high political priority. Community-based care may be supported but not given adequate resources.

Piat et al. [59] also suggested that implementing recovery guidelines is difficult for a number of reasons, including the perception among stakeholders that the guidelines themselves are complex, ambiguous or unclear. While working in Canada, these authors translated the recovery-oriented guide into an implementation strategy by forming implementation teams, comprising different stakeholders (service users, service providers, managers, knowledge users), and facilitating a 12-meeting implementation planning process. This strategy is a valuable model for a successful implementation program. However, we admit that our strategy to search the guideline as the primary source of the data analysis might have some limitations. For example, we might have missed out on some guidelines. Another limitation of this study is that we only reviewed the published guidelines written in English. We suggest that future studies should address these limitations.

## Conclusion

The "recovery-oriented" concept has been adopted as a new framework for mental health services globally. Most of the developing countries have released guidelines for this framework. Through a narrative synthesis, this research identified seven common recovery-oriented principles for mental health services by synthesizing the guidelines from five countries. These include cultivating hope, person-centeredness and empowerment, acknowledging an individual's uniqueness and social context, providing social support, establishing partnerships and collaborations, recognizing consumer's rights, and ensuring organizational commitment and evaluation of service delivery. They will be used as the guiding principles for our project which is based in the community health center of the Kulonprogo District, Special Province of Yogyakarta, Indonesia. Currently, the project is still in the early stage of implementation, and we are aware that implementing these guidelines is not an easy task. However, we have received a strong support from the local government in this regard; recovery-oriented services have been incorporated in the five-year action plan of the local government. We hope that this framework will be adopted by the central government in Indonesia and other developing countries.

## Acknowledgments

We would like to thank the District Health Office of the Kulonprogo District, Special Province of Yogyakarta for their continuous collaboration.

## Author Contributions

**Conceptualization:** M. A. Subandi, Carla R. Marchira, Trihayuning Tyas, Ariana Marastuti, Osi Kusuma Sari, Mary-Jo D. Good, Byron J. Good.

**Data curation:** Maryama Nihayah, Ratri Pratiwi, Fiddina Mediola, Yohanes K. Herdiyanto.

**Formal analysis:** M. A. Subandi, Ratri Pratiwi, Yohanes K. Herdiyanto.

**Funding acquisition:** M. A. Subandi, Carla R. Marchira, Trihayuning Tyas, Ariana Marastuti.

**Methodology:** Ariana Marastuti, Yohanes K. Herdiyanto, Osi Kusuma Sari.

**Project administration:** Maryama Nihayah, Ratri Pratiwi.

**Resources:** Maryama Nihayah, Trihayuning Tyas, Ratri Pratiwi, Fiddina Mediola, Osi Kusuma Sari.

**Supervision:** Carla R. Marchira, Mary-Jo D. Good, Byron J. Good.

**Writing – original draft:** M. A. Subandi, Maryama Nihayah, Ratri Pratiwi, Fiddina Mediola.

**Writing – review & editing:** M. A. Subandi, Byron J. Good.

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
