## [Decision Letter · Decision Letter 0]

18 Jul 2022

PONE-D-22-11468The principles of recovery-oriented mental health services: A review of the guidelines from five different countriesPLOS ONE

Dear Dr. MA Subandi,

Thank you for submitting your manuscript to PLOS ONE. After careful consideration, we feel that it has merit but does not fully meet PLOS ONE’s publication criteria as it currently stands. Therefore, we invite you to submit a revised version of the manuscript that addresses the points raised during the review process.

We look forward to receiving your revised manuscript.

Kind regards,

Rogis Baker, Ph.D

Academic Editor

PLOS ONE

Journal Requirements:

“This study was funded by Innovative and Productive Research Program, the Minister of Finance, the Government of Indonesia

The authors who received the award: MAS CM TT AM are

Grant numbers : 110/LPDP/2019

URL funder :https://lpdp.kemenkeu.go.id/riset/kebijakan-rispros-umum/”

Reviewers' comments:

Reviewer's Responses to Questions

**Comments to the Author**

1. Is the manuscript technically sound, and do the data support the conclusions?

Reviewer #1: Yes

Reviewer #2: Yes

2. Has the statistical analysis been performed appropriately and rigorously? 

Reviewer #1: Yes

Reviewer #2: Yes

3. Have the authors made all data underlying the findings in their manuscript fully available?

Reviewer #1: Yes

Reviewer #2: Yes

4. Is the manuscript presented in an intelligible fashion and written in standard English?

Reviewer #1: Yes

Reviewer #2: Yes

5. Review Comments to the Author

Reviewer #1: The paper is well organized and has a clear viewpoint, outlining 7 principles of recovery-oriented mental health services. However, we see that in Method, the inductive thematic analysis was conducted by multiple people, which may have errors in the generalization of different concepts. Although a joint theme review was subsequently conducted, it is not stated whether there was a uniform quality control by senior professionals to reduce the possible errors.

Reviewer #2: Thanks for asking me to review this manuscript titled: “The principles of recovery-oriented mental health services: A review of the guidelines from five different countries”

The authors have written a manuscript that summarizes the theoretical background to their attempt towards developing a guideline for establishing a recovery-oriented mental health service in Indonesia. Their decision to leverage what has been implemented in other countries establishes a firm theoretical foundation for their intended program. I have no reservations in recommending that their manuscript be published as this will form a guide for other developed countries who might desire to pursue the same course.

However, I have some concerns.

Title

The title does not seem to capture the essence of this manuscript. The manuscript isn’t just about a review. It is a review of guidelines for the development of a protocol. This is also seen in the conclusion of the abstract where the authors write “We will adjust and implement the result of the review in our project focusing on developing recovery-oriented mental health service in the community health centre in Yogyakarta, Indonesia. We hope that this framework will be adopted by the central government in Indonesia and other developing countries”. I suggest that the title should be modified to reflect what this manuscript is about.

Aim

The authors stated that the aim of the review is to guide the direction of the entire project. This doesn’t suffice as a specific aim of the manuscript. If the aim is to guide the direction of the entire project, then the title should actually be about the entire project. The authors need to determine the aim of the research, and let it be in tandem with the title.

Methods

The authors stated that “We did not use formal search engines, such as Scopus, EBSCO, or ProQuest, since they mostly search for journal articles that do not have practical guidelines.” Are they sure that they would not have missed out on some studies? This might need to be stated as a limitation. Restricting the publications to those written in English should also be stated as a limitation.

A chart showing how the publications reviewed were excluded before the final selection can also be helpful.

Results

Expatiating on the results from line 182 downwards looked more like discussions which needed to be cited. There is a need to rewrite this section.

Discussion

The aim needs to be properly phrased to make it be in tandem with the discussion as it is. Otherwise, the discussion will appear to be superfluous.

Many thanks for this opportunity.

6. PLOS authors have the option to publish the peer review history of their article (what does this mean?). If published, this will include your full peer review and any attached files.

Reviewer #1: No

Reviewer #2: **Yes: **Olanrewaju Ibikunle Ibigbami

---

## [Author Response · Author response to Decision Letter 0]

5 Oct 2022

Review Comments to the Author

Reviewer #1: The paper is well organized and has a clear viewpoint, outlining 7 principles of recovery-oriented mental health services. However, we see that in Method, the inductive thematic analysis was conducted by multiple people, which may have errors in the generalization of different concepts. Although a joint theme review was subsequently conducted, it is not stated whether there was a uniform quality control by senior professionals to reduce the possible errors.

Response:

We have added one sentence to follow this suggestion (lines 161-162)

Reviewer #2: Thanks for asking me to review this manuscript titled: “The principles of recovery-oriented mental health services: A review of the guidelines from five different countries”

The authors have written a manuscript that summarizes the theoretical background to their attempt towards developing a guideline for establishing a recovery-oriented mental health service in Indonesia. Their decision to leverage what has been implemented in other countries establishes a firm theoretical foundation for their intended program. I have no reservations in recommending that their manuscript be published as this will form a guide for other developed countries who might desire to pursue the same course.

However, I have some concerns.

Title

The title does not seem to capture the essence of this manuscript. The manuscript isn’t just about a review. It is a review of guidelines for the development of a protocol. This is also seen in the conclusion of the abstract where the authors write “We will adjust and implement the result of the review in our project focusing on developing recovery-oriented mental health service in the community health centre in Yogyakarta, Indonesia. We hope that this framework will be adopted by the central government in Indonesia and other developing countries”. I suggest that the title should be modified to reflect what this manuscript is about.

Response:

We have changed the title to follow this suggestion (lines 3-4)

Aim

The authors stated that the aim of the review is to guide the direction of the entire project. This doesn’t suffice as a specific aim of the manuscript. If the aim is to guide the direction of the entire project, then the title should actually be about the entire project. The authors need to determine the aim of the research, and let it be in tandem with the title.

Response:

We added some words so that the aim aligns with the title (lines 32-33).

Methods

The authors stated that “We did not use formal search engines, such as Scopus, EBSCO, or ProQuest, since they mostly search for journal articles that do not have practical guidelines.” Are they sure that they would not have missed out on some studies? This might need to be stated as a limitation. Restricting the publications to those written in English should also be stated as a limitation.

A chart showing how the publications reviewed were excluded before the final selection can also be helpful.

Response:

We added limitations of the study in the end of the Discussion section (lines 1016-1037)

Results

Expatiating on the results from line 182 downwards looked more like discussions which needed to be cited. There is a need to rewrite this section.

Discussion

The aim needs to be properly phrased to make it be in tandem with the discussion as it is. Otherwise, the discussion will appear to be superfluous.

Response:

We have made a significant change in the Result and Discussion sections. First, following the reviewers' suggestions, we checked the themes identified in the previous analysis by using the keywords of each theme and calculated the frequency of each theme in the guidelines. Based on this frequency, we could make an order of the list for the theme. Another significant change in the Result is that we explicitly mention and refer to the guidelines as the source of analysis. By doing this, the readers can differentiate the Result section (using References from the guidelines) from the Discussion session (using References from the literature).

---

## [Editor Report · Decision Letter 1]

14 Oct 2022

The principles of recovery-oriented mental health services: A review of the guidelines from five different countries for developing a protocol to be implemented in Yogyakarta, Indonesia.

PONE-D-22-11468R1

Dear Dr. MA Subandi,

We’re pleased to inform you that your manuscript has been judged scientifically suitable for publication and will be formally accepted for publication once it meets all outstanding technical requirements.

Kind regards,

Rogis Baker, Ph.D

Academic Editor

PLOS ONE